# Cross-plane coherent acoustic phonons in two-dimensional organic-inorganic hybrid perovskites

Peijun Guo[1], Constantinos C. Stoumpos [2], Lingling Mao[2], Sridhar Sadasivam[1], John B. Ketterson[3], Pierre Darancet[1], Mercouri G. Kanatzidis[2] & Richard D. Schaller[1,2]

Two-dimensional Ruddlesden–Popper organic–inorganic hybrid layered perovskites (2D RPs) are solution-grown semiconductors with prospective applications in next-generation optoelectronics. The heat-carrying, low-energy acoustic phonons, which are important for heat management of 2D RP-based devices, have remained unexplored. Here we report on the generation and propagation of coherent longitudinal acoustic phonons along the cross-plane direction of 2D RPs, following separate characterizations of below-bandgap refractive indices. Through experiments on single crystals of systematically varied perovskite layer thickness, we demonstrate significant reduction in both group velocity and propagation length of acoustic phonons in 2D RPs as compared to the three-dimensional methylammonium lead iodide counterpart. As borne out by a minimal coarse-grained model, these vibrational properties arise from a large acoustic impedance mismatch between the alternating layers of perovskite sheets and bulky organic cations. Our results inform on thermal transport in highly impedance-mismatched crystal sub-lattices and provide insights towards design of materials that exhibit highly anisotropic thermal dissipation properties.

[1] Center for Nanoscale Materials, Argonne National Laboratory, 9700 South Cass Avenue, Lemont, IL 60439, USA. [2] Department of Chemistry, Northwestern University, 2145 Sheridan Road, Evanston, IL 60208, USA. [3] Department of Physics and Astronomy, Northwestern University, 2145 Sheridan Road, Evanston, IL 60208, USA. Correspondence and requests for materials should be addressed to R.D.S. (email: schaller@anl.gov)

Hybrid organic–inorganic perovskites such as methylammonium lead halides have emerged as extraordinary light absorbing and emitting materials[1,2]. Improved understanding of the unique features such as extended carrier lifetime, hot-carrier protection, and dynamic disorder have been advanced by innovative theoretical and experimental approaches[3–7]. Two-dimensional (2D) Ruddlesden–Popper organic–inorganic hybrid perovskites (2D RPs)[8–10], as alternatives to the 3D counterparts, have gained significant attention owing to improved ambient stability[11], high luminescence quantum yield[12] and strong excitonic effects[13,14]. The 2D RPs consist of sheets of corner-sharing Pb–X (X = I, Br) octahedra expanding in two dimensions, separated by interdigitating bilayers of organic cations, forming a highly hierarchical architecture. While intensive efforts have revealed their attractive properties such as enhanced optical nonlinearity[15], strong electron–phonon interaction[16,17], and white-light emission[18], no studies so far have addressed phonon transport, a crucial aspect for proper thermal management of 2D RP-based devices. In particular, the interlocked soft organic and hard inorganic sub-lattices raise intriguing questions regarding the existence and properties of low-energy vibrational excitations, and their ability to carry heat along the cross-plane direction of these highly anisotropic materials with natural quantum-well-like electronic structures.

2D RPs additionally present features of superlattices, an attractive material class that can exhibit desirable thermal properties that differ from the bulk limits[19] while still preserving the wave nature of phonons[20,21]. While solution-synthesized 2D RPs can be viewed as superlattices, they qualitatively differ from the engineered all-inorganic versions (e.g., Si–Ge[22] and AlAs–GaAs[23]) due to their atomically abrupt interfaces with minimal misfit dislocations or interfacial atomic mixing, a high degree of long-range order, and, most importantly, the extreme heterogeneity of the atomic masses of the organic and inorganic sub-lattices connected in a serial fashion. Moreover, the van der Waals bonding that holds the structure of 2D RPs together closely resembles the interlayer interactions in 2D van der Waals heterostructures[24] and MXenes[25]. Yet, unlike those analogs, 2D RPs can be grown into single crystals with tens-to-hundreds-of-micron cross-plane dimensions, offering an archetypal system for the study of intrinsic phonon transport characteristics.

Here we investigate coherent longitudinal acoustic phonons (CLAPs) in a series of 2D RPs, $(CH_3(CH_2)_3NH_3)_2(CH_3NH_3)_{N-1}Pb_NI_{3N+1}$, or $(BA)_2(MA)_{N-1}Pb_NI_{3N+1}$ ($N = 1$–6), which are denoted as $N = 1$–6 for short. Single-crystal transient reflection measurements, aided by the extracted indices of refraction, permit direct access to the cross-plane acoustic phonon group velocities with high fidelity. We find 2D RPs exhibit significantly lower group velocities compared to the 3D counterpart, methylammonium lead iodide (MAPbI$_3$), which corresponds to $N = \infty$. As captured by a coarse-grained bead-spring model, the reduction in phonon velocity originates from the weak van der Waals bonding between the organic spacers. The unusually large acoustic impedance mismatch between the organic and inorganic layers, and the loose $CH_3(CH_2)_m$ aliphatic tails of the organic spacer layers with partial liquid-like motions lead to substantially stronger suppression of acoustic phonon propagation in 2D RPs as compared to MAPbI$_3$. We expect our findings based on such model material class to be applicable to a broader class of systems, such as nanocrystal arrays[26], biological organic–inorganic interfaces[27], and self-assembled monolayers on inorganic solids[28,29].

## Results

### Structural characterization of the 2D RPs.
The synthesis of 2D RPs reported here followed the previously reported procedures[10,30]. Figure 1a shows the unit cell of $N = 4$ as a representative example of the crystal structure. The 2D RPs grow as single-crystal flakes with orthorhombic structures, and have a rectangular parallelepiped crystal habit. We designate $[(MA)_{N-1}Pb_NI_{3N+1}]^{2-}$ as the inorganic layer (although it contains MA$^+$ cations that are ionically bound to the inorganic lattice), which is comprised of $N$ connected sheets of Pb–I octahedra (Fig. 1a). Each $[(BA)_2]^{2+}$ layer, comprised of two van der Waals-bonded BA$^+$ sheets, is referred to as the organic layer. The characteristic $\theta$–$2\theta$ X-ray diffraction peaks (Fig. 1b and Supplementary Table 1) show that the single-crystal flakes have an out-of-plane direction along the $b$ axis, and lateral dimensions coincident with the Pb–I sheets in the $a$–$c$ plane. Higher-order diffraction peaks (Supplementary Fig. 1) and photoluminescence spectra (Fig. 1c) together reveal that each crystal is composed of a single phase, including the high members ($N = 5$ and 6) that were thought to be difficult to isolate in pure form[31].

### Transient reflection measurements on the 2D RPs.
We performed transient reflection measurements using 400-nm (above-bandgap) pumping and near-infrared probing at nearly normal incidence on the $a$–$c$ plane of the single-crystal flakes (Fig. 1a). The above-bandgap optical pumping excites hot electron–hole pairs, which impulsively heat up the lattice through electron–phonon coupling and launch the CLAPs[32,33]. The pump-induced heating of the crystal surface produces cross-plane propagating CLAPs (along the $b$ axis) and with it a local, strain-induced refractive index modulation. This propagating lattice wave packet with a locally modulated index reflects the probe

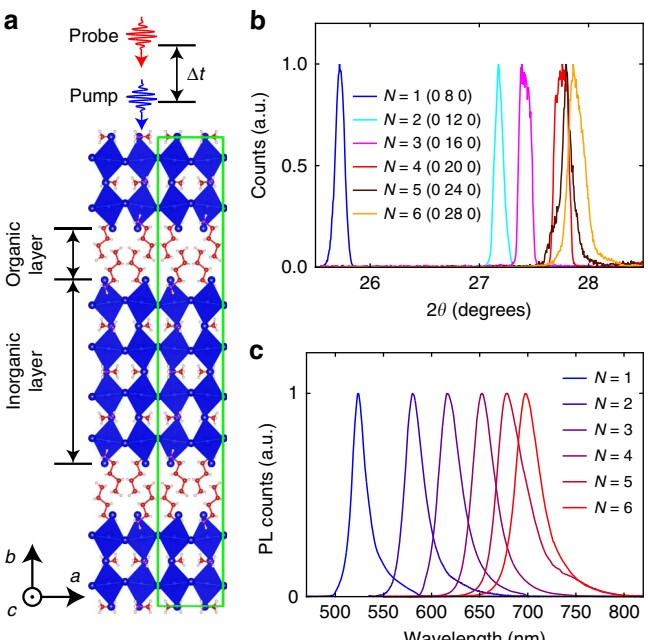

**Fig. 1** The crystal structure and phase purities of 2D RPs. **a** The crystal structure of $N = 4$ with a unit cell enclosed by the green rectangle. The directions of the pump and probe beams with respect to the crystal orientation are illustrated on the top. **b** Characteristic X-ray diffraction peaks of the single-crystal flakes ($N = 1$–6) obtained from $\theta$–$2\theta$ scans. **c** Photoluminescence spectra of the 2D RPs

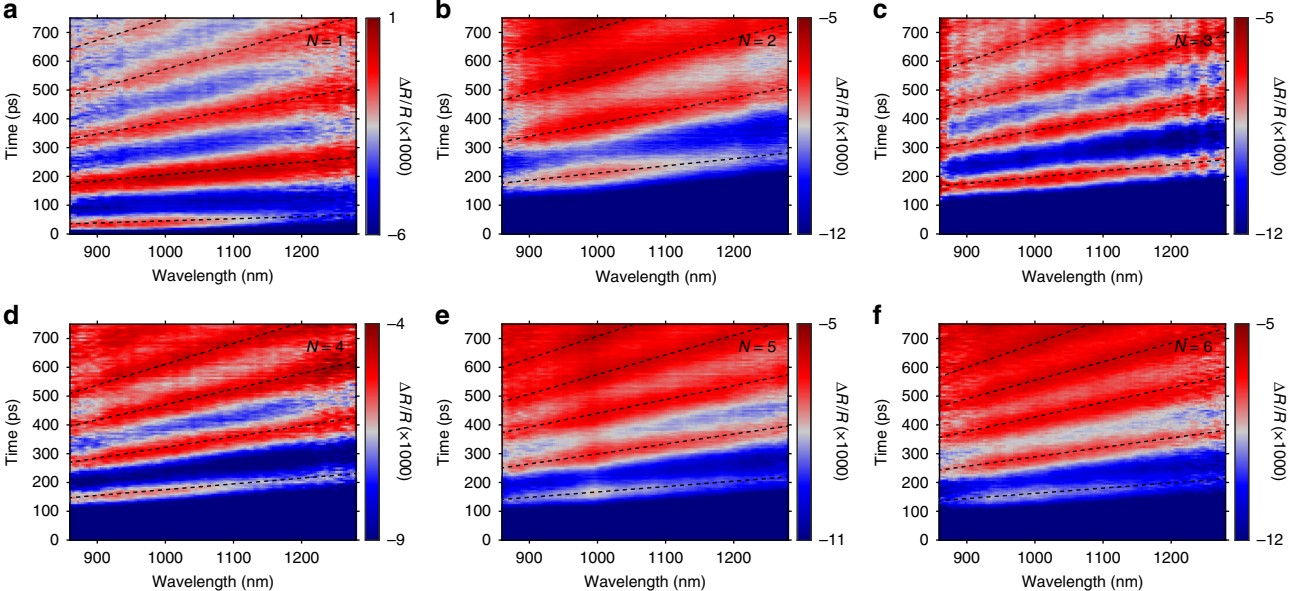

**Fig. 2** Transient reflection measurements of the 2D RP single crystals. **a–e** $\Delta R/R$ spectral maps for $N = 1$–6, respectively, acquired with 400-nm pump excitation. The scalebar is in unit of $\Delta R/R$ (×1000). Fluences used were 0.18, 0.25, 0.25, 0.25, 0.18, and 0.21 mJ cm$^{-2}$, respectively

light, which interferes with the probe light reflected from the front surface of the crystal, producing time-dependent oscillations in the probe intensity. Figure 2 presents transient $\Delta R/R$ spectral maps acquired for $N = 1$–6 for probe wavelengths between 860 and 1280 nm. The color-coded $\Delta R/R$ is the differential change of reflection, defined as $[R(t) - R_0]/R_0$ with $R_0$ being the reflection with pump off and $R(t)$ with pump on (at delay time $t$). We found that probing in the below-bandgap range (where the penetration depth is large) is necessary for the detection of CLAPs, which propagate microns deep into the crystal. The period $T$ of the probe oscillation is related to the CLAP velocity $v$ through $T(\lambda) = \frac{\lambda}{2vn'(\lambda)} a$, where $n'(\lambda)$ is the real part of refractive index at wavelength $\lambda$. Each period (in time) of the oscillation corresponds to a phonon propagation distance equal to half the wavelength of light in 2D RPs. As a result, the propagation distances of the CLAPs shown in Fig. 2 are on the order of a few microns.

**Characterization of the refractive indices for the 2D RPs.** Calculation of $v$ based on $T(\lambda)$ obtained from Fig. 2 requires knowledge of $n'(\lambda)$ in the below-bandgap range, which has not been reported for 2D RPs. Here we developed a two-step procedure to determine $n'(\lambda)$. We first measured the below-bandgap reflection spectra of a-few-μm thick single-crystal flakes. Because near-infrared light fully penetrates the sample, interferences owing to reflection from the top (air–material) and bottom (material–air) interfaces yield a series of maxima and minima in the reflection spectrum, at the respective conditions of $2n'(\lambda)l = (m - 1/2)\lambda$ and $2n'(\lambda)l = m\lambda$, where $l$ is the sample thickness and $m$ is a positive integer denoted as a mode number (note that each pair of adjacent reflection minimum and maximum has a specific mode number). Figure 3a shows a scanning electron microscopy (SEM) image of an $N = 6$ flake with a smooth surface. A thickness for this region of 6.29 μm follows from the side-view image (Fig. 3b). The reflection spectrum for this same region (Fig. 3c) exhibits several minima, which were assigned to different mode numbers. With increasing wavelength, $n'(\lambda)$ decreases monotonically below the bandgap (as dictated by the Kramers–Kronig relation) and with it the mode number $m = \frac{2n'(\lambda)l}{\lambda}$. For two neighboring reflection minima, $m$ should

differ by one, and assigning a mode number to the bluest minimum (denoted as $m_0$) fixes those for the other minima, which subsequently permits the calculation of $n'(\lambda)$ over the range where the reflection minima are observed, via the equation $2n'(\lambda)l = m\lambda$. As shown in Fig. 3d, using different trial values for $m_0$ we can produce several $n'(\lambda)$ curves, only one of which is correct. To resolve this ambiguity, a separate measurement was performed to determine $n'(\lambda)$ at the absorption edge (where reflection from the bottom interface is suppressed due to light absorption), which is only slightly bluer than the bluest reflection minimum. The detailed procedure appears in Supplementary Note 1 and Supplementary Figs. 2–11. This value of $n'(\lambda)$, shown as the cyan dot in Fig. 3d, permits the identification of the correct $n'(\lambda)$ curve, as shown in red. We repeated the same procedure for $N = 5$–1, with the corresponding results shown in Supplementary Figs. 2–7. Our accurate determination of $n'(\lambda)$ leverages the strong interference effect, which is attributable to the uniform thickness of the single-crystal flakes, a direct result of the layer-by-layer growth unique to the 2D RPs (Supplementary Fig. 12).

**CLAP velocities of the 2D RPs.** The $n'(\lambda)$ curves for $N = 1$–6 together with the reported[34] $n'(\lambda)$ for MAPbI$_3$ appear in Fig. 4a. As expected, $n'(\lambda)$ decreases with increasing $\lambda$ (a larger spectral separation from the bandgap, $E_g$). Furthermore, $n'(\lambda)$ decreases with a decreasing $N$, which stems from a larger volumetric ratio of the organic sub-lattice that has lower dielectric constant than the inorganic counterpart. The variation of $n'(\lambda)$ with $E_g$ is qualitatively consistent with the Moss relation[35], written as $\left[n'\left(E_g\right)\right]^4 E_g = 95$ eV. With the determined $n'(\lambda)$, and the CLAP oscillation period $T(\lambda)$ taken as time difference between the consecutive maxima in the $\Delta R/R$ map (averaged over two periods; black dashed lines in Fig. 2), we calculated $v$ and plot it against $N$ in Fig. 4b. Note that from $T(\lambda) = \frac{\lambda}{2vn'(\lambda)}$, $v$ can be calculated using $T(\lambda)$ and $n'(\lambda)$ at each wavelength, and should be consistent throughout the entire probed window. Figure 4b plots the mean value and standard deviation of $v$ calculated using the $\Delta R/R$ map from 860 to 1050 nm, where both $T(\lambda)$ and $n'(\lambda)$ are known. The variation in $v$ within the probe wavelength window is as low as

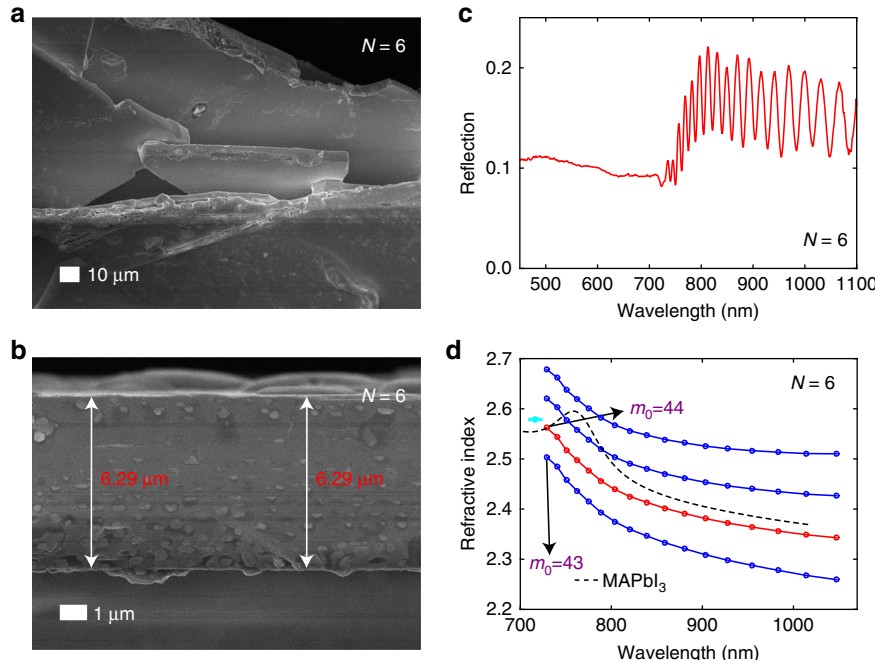

**Fig. 3** Determination of the refractive index for $N = 6$. **a** Angular and **b** side-view of the examined flake taken under SEM. **c** Reflection spectrum and **d** several possible $n'(\lambda)$ curves obtained using different $m_0$ values, with the correct $n'(\lambda)$ curve shown in red. The cyan dot denotes the refractive index at the absorption onset wavelength determined from a separate step

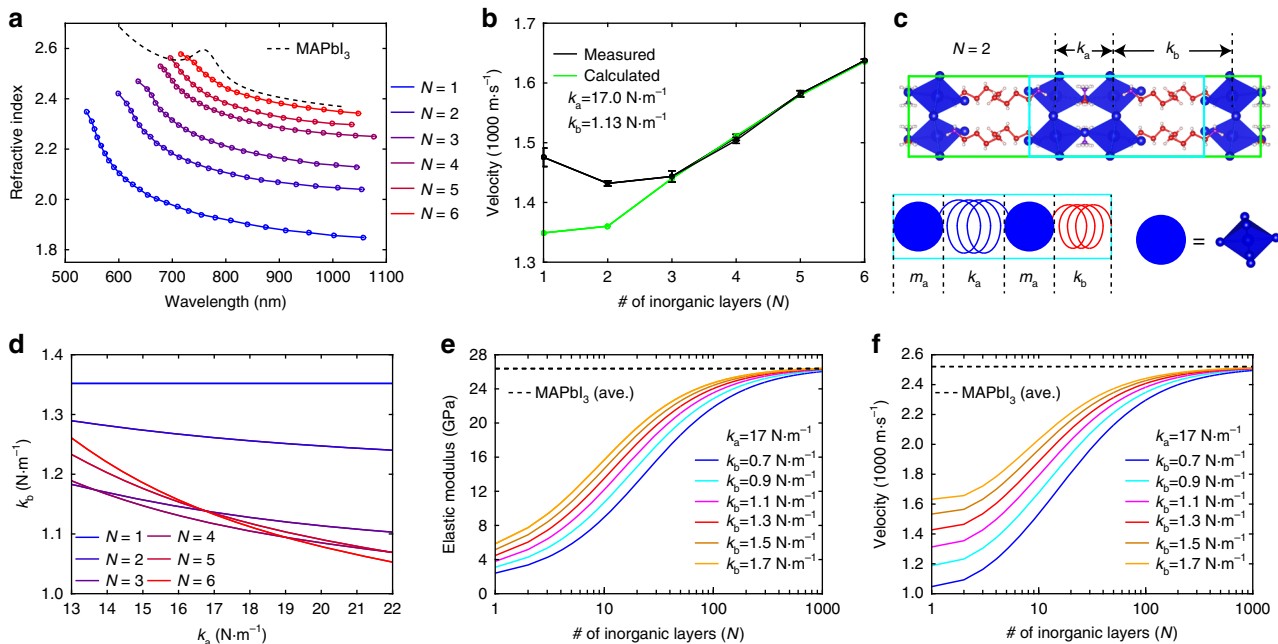

**Fig. 4** The CLAP velocities of 2D RPs. **a** $n'(\lambda)$ curves for $N = 1–6$ in the below-bandgap range, and $n'(\lambda)$ for $N = \infty$ taken from literature. **b** Measured CLAP velocities plotted against $N$ (number of inorganic layers), shown in black. Calculated results from the bead-spring model is shown in green. **c** Schematic illustration of the coarse-grained bead-spring model. In the upper figure, the green rectangle encloses a single unit cell, half of which, highlighted by a cyan rectangle, was considered in the bead-spring model, as illustrated in the lower figure. **d** The pairs of the spring constants that give the calculated velocities exactly matching the experimental ones for $N = 1–6$. Dependence of elastic modulus along the cross-plane direction in **e** and velocity in **f** on the number of inorganic layers plotted for various $k_b$ values (with $k_a$ fixed at 17 N m$^{-1}$). Result for MAPbI$_3$ (average value of its lower and upper bounds) is shown in black dashed lines

0.7% (averaged for $N = 1–6$), which confirms the accurate determination of $n'(\lambda)$. Three interesting observations arise from Fig. 4b. (i) For $N = 1–6$, we find $v$ to be within 1430–1640 m s$^{-1}$, which is significantly lower than that for $N = \infty$ (between 2340 m s

$^{-1}$ and 2990 m s$^{-1}$, as discussed later). (ii) Decreasing $N$ from 6 to 2 causes a reduction of $v$, with the exception of $v$ for $N = 1$ which falls in between that of $N = 2$ and $N = 3$. (iii) Varying $N$ from 1 to 6, which lowers the weight percentage of the organic layer by

six times (from 20.75% reduced to 3.89%), does not strongly impact $v$ (which only varies up to 14.4%). The measured values of $v$ for $N = 1$–$6$ are 1476, 1432, 1444, 1495, 1582 and 1637 m s$^{-1}$, respectively.

**Coarse-grained bead-spring model**. The substantial reduction of $v$ upon the periodic inclusion of organic layers in a parent MAPbI$_3$ lattice arises from the distinctively different bond strengths and atomic masses of the two species. The acoustic impedances, defined as $\rho v$, where $\rho$ and $v$ are the mass density and sound velocity of the constituents, are estimated to be $1.11 \times 10^7$ kg m$^{-2}$ s$^{-1}$ and $9.25 \times 10^5$ kg m$^{-2}$ s$^{-1}$ for the respective inorganic and organic layers. Such a large impedance mismatch can cause appreciable reduction in CLAP velocity in the composite[36]. Here $\rho$ and $v$ of liquid *n*-butylamine (0.74 g cm$^{-3}$ and 1249.8 m s$^{-1}$) were assumed for the organic layer[37]; note that the organic layer in a periodic arrangement can exhibit larger sound velocity than liquid phase of butylamine with random molecular orientations (as true for the smectic-A phase compared to the isotropic phase for liquid crystals[38]). To gain a deeper understanding, we developed a minimal coarse-grained, one-dimensional (1D) bead-spring model (schematically illustrated in Fig. 4c for the case of $N = 2$). Such a 1D model is sufficient to capture the essence of the long-wavelength phonon propagation along the cross-plane direction, which is the experimentally measured quantity. Details of the bead-spring model appears in Supplementary Note 2 and Supplementary Fig. 13. First-principles calculations of the phonon band structures were not attempted because of the very large unit cells of the high $N$ members, and the local octahedral distortions and dynamic disorder in lead-halide perovskites, which often result in imaginary modes in the acoustic branches[39]. In the 1D bead-spring model, we represent each Pb–I octahedral sheet by a blue bead, which has the mass ($m_a$) of one Pb–I octahedron. Two neighboring Pb–I octahedra in direct contact are connected by a blue spring with spring constant $k_a$; two Pb–I octahedra separated by the organic layer are linked by a red spring with spring constant $k_b$. The blue spring has a length of 6.39 Å, which is the average distance between adjacent octahedral sheets in $N = 1$–$6$ (which varies within 2.54%). The red spring has a length of 13.24 Å, which is about 7 Å longer than the blue spring, due to the inclusion of the organic layer.

To get a reasonable estimate of $k_a$, we computed the CLAP velocity for a chain of blue beads connected by $k_a$, which we denote as $v_a$. Because the octahedral sheets in 2D RPs derive from the (110) planes of the tetragonal MAPbI$_3$ (plane spacing is 6.26 Å), in the sense that the perovskite layers in both cases are stacked in an eclipsed manner (Supplementary Fig. 14), $v_a$ should be approximately equal to the velocity along the [110] direction of the tetragonal phase, which transforms into the [100] direction of the cubic phase upon the phase transition[40]. Earlier measurements on MAPbX$_3$ single crystals yielded a velocity of 2340 m s$^{-1}$ along the [100] direction of the tetragonal MAPbI$_3$[41], corresponding to the [110] direction for the cubic phase. Hence, the velocity along the [100] direction of the cubic MAPbI$_3$, and with it $v_a$, should be greater than 2340 m s$^{-1}$, since cubic MAPbBr$_3$ is stiffer along the [100] than the [110] direction[42], which we assume also holds for MAPbI$_3$. We also measured a velocity of 2990 m s$^{-1}$ along the [100] direction of the cubic MAPbBr$_3$, which is stiffer than cubic MAPbI$_3$ due to a stronger Pb–Br bond. Therefore, $v_a$ should lie between 2340 m s$^{-1}$ and 2990 m s$^{-1}$, which results in an estimated value for $k_a$ within the 13.0–21.0 N m$^{-1}$ range (Supplementary Fig. 15).

The equations of motion, considering nearest-neighbor interactions (Supplementary Note 2), result in a set of linear equations,

where solving for the low frequency limit yields $v$. We found that by fixing $k_a = 17$ N m$^{-1}$ (the average of its lower and upper bound), the experimental results for $N = 4$–$6$ are well reproduced with $k_b = 1.13$ N m$^{-1}$ (Fig. 4b). We note that the effect of Pb–I octahedral distortions on the acoustic phonon velocity along the cross-plane direction is implicitly incorporated into the effective spring constant ($k_a$) that connects the beads. By fixing the value of $k_a$, the differences in octahedral rotations in $N = 1$–$6$ are implicitly neglected, because we found[41] that octahedral rotations, which occurred in MAPbI$_3$ as temperature is varied from 300 K to 160 K, only results in a minor change of velocity within 10%. Notably, the more than one order of magnitude smaller $k_b$ compared to $k_a$ possibly arises from (i) the weak van der Waals interactions between the CH$_3$(CH$_2$)$_m$ tails of the BA$^+$ cations; (ii) the weak vibrational coupling between the Pb–I octahedron and the amine groups of the BA$^+$ cation. If we fix $k_a = 17$ N m$^{-1}$ and $k_b = 1.13$ N m$^{-1}$, the calculated $v$ can match the measured values for $N = 4$–$6$, but progressively underestimates $v$ as $N$ further decreases. This can be attributed to an increasing ionic character of the inorganic layer as $N$ decreases from 6 to 1. In particular, because $N$ layers of perovskite, with the chemical formula [(MA)$_{N-1}$Pb$_N$I$_{3N+1}$]$^{2-}$, always carry $-2$ charges to compensate the $+2$ charges carried by (BA)$_2^{2+}$, each perovskite layer on average distributes $-2/N$ charge. As a result, in lower $N$ members, the more negatively charged inorganic layer is more strongly attracted to the positively charged BA$^+$ via Coulomb interactions. Additional calculations were performed in which $k_b$ was adjusted to match the velocity for each $N$, when we swept $k_a$ within its lower and upper bounds. As shown in Fig. 4d, for a fixed value of $k_a$ within its lower and upper bounds, $k_b$ for the lower $N$ members can be 10–30% larger than that for higher $N$ members. For any given member, the small variation of $k_b$ (<10%) relative to the variation of $k_a$ (13–22 N m$^{-1}$) stems from the order-of-magnitude difference between $k_b$ and $k_a$; such difference dictates that the CLAP velocity is mainly determined by $k_b$, while a relatively large variation of $k_a$ does not significantly alter the velocity.

Using the bead-spring model, we further examined the asymptotic behavior of $N > 6$. The lattice constants along $b$ were extrapolated from those of $N = 1$–$6$ (Supplementary Fig. 16). We find each additional inorganic layer on average increases $b$ by 6.203 Å, whereas the average thickness of the organic layer is 7.437 Å. We first calculated the diagonal components of the elastic modulus along the cross-plane direction, denoted as $E$. Fig. 4e shows the calculated $E$ as a function of $N$ obtained with various $k_b$ values, wherein we fix $k_a$ at 17 N m$^{-1}$. The velocity, shown in Fig. 4f, was then evaluated using $v = (E/\rho)^{1/2}$ where $\rho$ is the mass density. Although $v$ approaches the MAPbI$_3$ limit for large $N$ values, substantial reduction in $v$ can be achieved even beyond $N = 6$ due to the soft and hard springs connected in series, whereas the optoelectronic properties (e.g., $E_g$ and refractive index) reach the MAPbI$_3$ limit much faster with $N$. Compared to $v$, a more dramatic reduction in $E$ is obtained, especially for the low $N$ members, partly due to the smaller $\rho$. Note that the lowest members ($N = 1$ and 2) have moduli about five times lower than that of MAPbI$_3$, and are comparable to those of common organic semiconductors (in the few-GPa range), with potential utility in flexible optoelectronic devices[43]. On the other hand, synthesis of higher 2D RP members ($N > 6$) may be possible, since pure phases of oxide RP perovskites[44] exist up to $N = 10$.

**Changing the organic spacers**. To investigate the origin of the small $k_b$ in comparison to $k_a$, we examined another 2D hybrid perovskite, (HA)PbI$_4$, where HA$^{2+}$ (histammonium, or

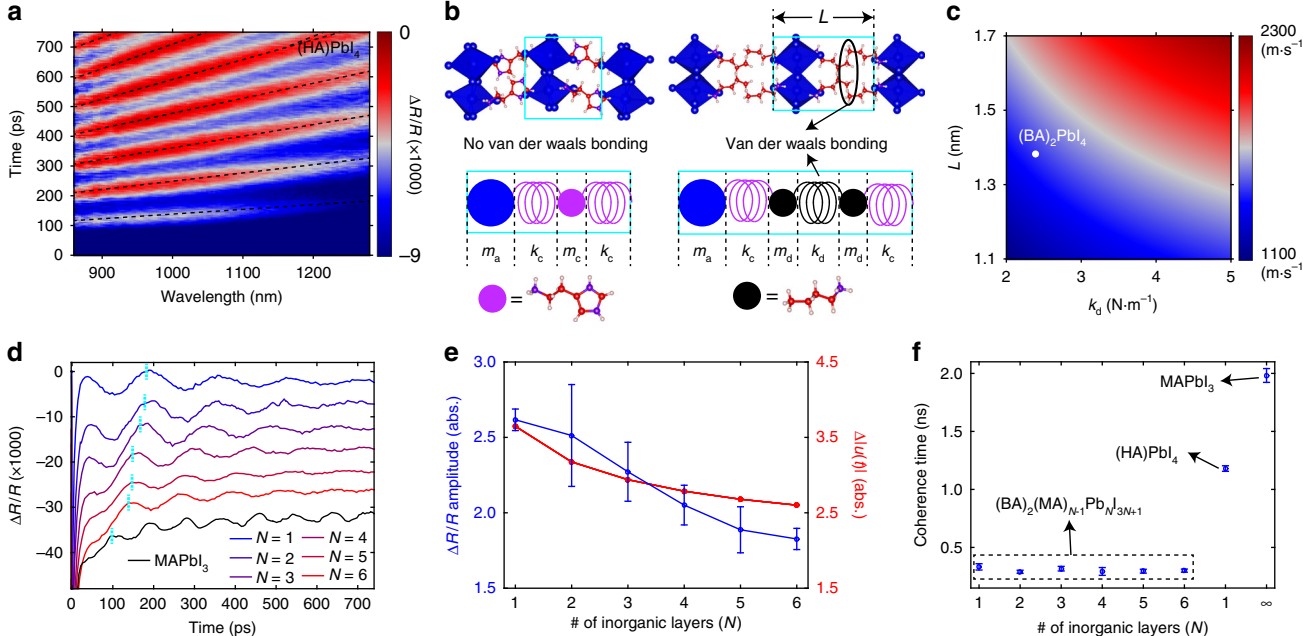

**Fig. 5** CLAP of (HA)PbI$_4$, and the amplitude and coherence time of CLAPs in 2D RPs and MAPbI$_3$. **a** $\Delta R/R$ spectral map for (HA)PbI$_4$ single-crystal flake, excited with 400-nm pump at a fluence of 0.2 mJ cm$^{-2}$. **b** Left: crystal structure of (HA)PbI$_4$ (upper figure) and schematic of the corresponding bead-spring model (lower figure). The ionic interaction between the Pb-I octahedron (blue bead) and the HA$^{2+}$ ion (purple bead) is represented by the purple spring with a spring constant of $k_c$. Right: crystal structure of $N = 1$ (upper figure) and schematic of the corresponding bead-spring model (lower figure). The van der Waals interaction between the two BA$^+$ ions (black beads) is represented by the black spring with a spring constant of $k_d$. **c** Dependence of the acoustic phonon velocity of $N = 1$ on $k_d$ and $L$, with $L$ defined in **b**. **d** $\Delta R/R$ kinetic traces at 955 nm for $N = 1$–6 and MAPbI$_3$ (curves are offset for clarify). **e** Blue: amplitude of the sinusoidal oscillations at the delay times associated with the first $\Delta R/R$ peak (marked by cyan dotted lines in **d**); the plotted amplitudes are normalized by the fluence described in caption of Fig. 2. Red: calculated time-averaged mean displacement of the beads. **f** Coherence time of the CLAP oscillations. Error bars (i.e., standard deviations) in **e** and **f** are based on the fitting of $\Delta R/R$ kinetics at ten different wavelengths spanning the probed spectral window

C$_3$N$_2$H$_4$CH$_2$CH$_2$NH$_3$$^{2+}$) is a dication that separates the individual perovskite layers[45]. The key distinction between (HA)PbI$_4$ and (BA)$_2$PbI$_4$ (which is denoted as $N = 1$), is that in (HA)PbI$_4$ two perovskite layers are connected by only one layer of HA$^{2+}$. As a result, (HA)PbI$_4$ does not have van der Waals bonding that is present in (BA)$_2$PbI$_4$. The transient $\Delta R/R$ map (Fig. 5a) together with the out-of-plane $n'(\lambda)$ for (HA)PbI$_4$ (Supplementary Figs. 17 and 18) obtained with a slightly different method (Supplementary Note 3), permit the determination of its cross-plane acoustic phonon velocity to be 1939 m s$^{-1}$. This value, notably, is 31% higher than the velocity of $N = 1$ (1476 m s$^{-1}$). To understand this significant difference in the velocities between (HA)PbI$_4$ and $N = 1$, we built another set of coarse-grained models (Fig. 5b), wherein the organic cations are explicitly included ($m_c$ and $m_d$ are the masses of HA$^{2+}$ and BA$^+$, respectively). The ionic bonding between the amine group and the Pb–I octahedron is represented by a purple spring with spring constant $k_c$, and the van der Waals bonding between the alphatic tails of the BA$^+$ cations is described by a black spring with spring constant $k_d$. Using the velocity for (HA)PbI$_4$, we calculated $k_c$ to be 10.3 N m$^{-1}$, which in turn permitted the estimation of $k_d$ to be 2.4 N m$^{-1}$. The significant difference between $k_c$ and $k_d$ clearly demonstrates that the slow cross-plane acoustic phonon velocities observed for $N = 1$ to $N = 6$ primarily stem from the weak van der Waals interactions between the organic spacer layers, whereas the organic–inorganic interface only plays a secondary role. Based on the model for $N = 1$, we further evaluated the effect of changing $k_d$ and $L$ (the total length of the inorganic and organic layer; see Fig. 5b) on its phonon velocity (Fig. 5c). Although it is expected that increasing $k_d$ can significantly increase the velocity, we find that a faster velocity

can also be achieved by using a longer organic spacer, through the reduction of the line density of the van der Waals bonds.

**Light-induced lattice dynamics.** Figure 5d shows the representative $\Delta R/R$ kinetic traces at 955 nm for $N = 1$–6, and reveals an increasing amplitude of the sinusoidal CLAP oscillation with decreasing $N$. This is quantitatively illustrated in Fig. 5e, which shows the fitted amplitude of the $\Delta R/R$ oscillation at the delay time of the first $\Delta R/R$ peak (cyan dashed lines in Fig. 5d), averaged over ten probe wavelengths spanning the monitored spectral window (Supplementary Fig. 19). To gain some understanding of this feature, we calculated $\Delta\langle|u(t)|\rangle$, the change of mean bead displacement due to energy injection into the system, using the bead-spring model (Supplementary Note 4 and Supplementary Figs. 20 and 21), which is again plotted in Fig. 5e. A similar dependence of $\Delta\langle|u(t)|\rangle$ on $N$ is observed, which may contribute to the observed $N$-dependent oscillation amplitude. Pump wavelength and fluence dependent measurements on $N = 2$ (strongly quantum confined) and $N = 4$ (weakly quantum confined), as shown in Supplementary Figs. 22–24, demonstrate that in both cases, the CLAP signatures in the $\Delta R/R$ spectral maps can only be obtained via above-bandgap pumping, but not with near-bandgap pumping. Therefore, the thermoelastic effect[46], as opposed to deformation potential, is mainly responsible for the CLAP signatures in the transient reflection spectral maps. Fully quantitative assessment of the $\Delta R/R$ oscillation amplitude requires knowledge of the photoelastic coefficients (i.e., strain-dependent refractive index change), which is beyond the scope of the present work. Note that the observed variation of oscillation amplitude with $N$ does not mainly arise from a difference in the

magnitude of index modulation due to a bandgap change (caused by CLAP-induced strain). Our first-principles calculation of the electronic band structure (Supplementary Fig. 25) shows a reduction in bandgap of 0.01 eV and an increase in bandgap of 0.04 eV for $N = 1$ and $N = 3$, respectively, under the application of 1% strain along the $b$ axis. A larger bandgap modulation achieved for the higher members is opposite in trend to the observed variation of the probe oscillation amplitude with $N$, indicating a negligible contribution from bandgap modifications to the refractive index change. Overall, changes of the band structure are small since the strain is absorbed by the soft organic layer without significant rearrangement of the inorganic sub-lattice; the latter encompasses the optoelectronically relevant conduction and valence bands.

## Discussion

The identified CLAP velocities have profound implications on the thermal conductivities ($\kappa$) of 2D RPs. In the kinetic theory, $\kappa$ is approximated as $\kappa = \frac{1}{3}Cv\Lambda$, where $C$ is the heat capacity and $\Lambda$ is the phonon mean free path. The small $v$ is expected to reduce $\kappa$ of 2D RPs below that of the MAPbI$_3$ value, which is already low (0.3–0.5 W m$^{-1}$ K$^{-1}$ as reported elsewhere[47]). Additionally, as plotted in Fig. 5f, the probe oscillations of $N = 1$–6 exhibit about six times shorter coherence times as compared to MAPbI$_3$ obtained from exponential fits (Supplementary Fig. 19), signifying a substantially increased attenuation of the CLAPs. Interestingly, we found that (HA)PbI$_4$ exhibits about four times longer coherence time than $N = 1$–6. This suggests that that dynamic disorders and large structural fluctuations of the organic spacer molecules at the aliphatic ends, which may undertake liquid motions due to the weak van der Waals interactions, plays an important role in impeding acoustic phonon transport. Strong anharmonicity at the organic–inorganic interfaces stemming from an asymmetric potential landscape may also contribute to phonon scattering (e.g., stronger interfacial reflection and phonon–phonon interactions)[48] and suppress cross-plane phonon propagation[49,50], which results in the overall short coherence times of the 2D RPs in comparison to 3D MAPbI$_3$. Note that measurements in the below-bandgap range with negligible optical absorption (Supplementary Fig. 11) ensure that optical absorption does not contribute to the observed acoustic phonon attenuation. For $N=1$–6, a seemingly independence of the coherence time on $N$ may arise from (i) larger defect densities in higher $N$ members which may contribute to additional acoustic phonon scattering[14]; (ii) a stronger manifestation of the wave-like nature of CLAPs in lower $N$ members with a higher interfacial density[20]; (iii) lower $N$ member exhibit stronger bonding strength at the organic interface (Fig. 4d), which possibly results in a less pronounced dynamic disorder. More quantitative understanding may be aided by measurements of the phonon mean free paths[23,51]. Based on the stronger CLAP attenuation, we expect $\kappa$ of 2D RPs to diminish more strongly than $v$ itself, and heat dissipation of 2D RP-based devices presents an important design consideration. Relatedly, it was shown that superlattices present low thermal conductivities along both the cross-plane and in-plane directions[52]; in the latter case phonon transport is extremely sensitive to the strengths of interfacial specular and diffuse scattering[53]. Further exploration of acoustic phonon propagations along the in-plane directions (which have superior charge transport properties) is warranted, and can be enabled by spatially resolved pump-probe techniques[54].

In conclusion, we find reduced group velocities and substantially stronger attenuation of acoustic phonons in 2D RPs as compared to MAPbI$_3$, which arises from the large impedance mismatch between the organic and inorganic layers due to the weak van der Waals bonding between the layers. 2D perovskites without van der Waals bonding (e.g., (HA)PbI$_4$) are expected to exhibit higher thermal conductivities for device applications, yet still maintaining strong quantum confinement effects. The demonstrated optical pump-probe technique is particularly powerful for the examination of acoustic phonons in strongly phonon absorptive (or reflective) media at reduced dimensions, where traditional ultrasonic methods[38] become problematic. Through a combination of atomic layer deposition and molecular layer deposition, both of which are layer-by-layer growth techniques with digital thickness control[55,56], hybrid organic–inorganic superlattices with arbitrary configurations are possible. These layered organic–inorganic hybrids feature an interesting class of materials with tunable acoustic phonon transport characteristics via chemical substitutions[57,58], and controllable distortions of the inorganic sub-lattice with corrugated structures[59,60].

## Methods

**Chemicals**. All chemicals were purchased from Sigma-Aldrich and used as received. Methylammonium iodide was synthesized by neutralizing equimolar amounts of 57% w/w aqueous hydriodic acid (HI) and 40% w/w aqueous methylamine (CH$_3$NH$_2$). The white precipitate was harvested by evaporating the solvent using a rotary evaporator at 60 °C under reduced pressure.

**Materials synthesis**. The 2D RPs were synthesized following the reported literature procedure[10,45], in a 20 mmol basis derived from the total Pb content. In a typical procedure, PbO powder (4464 mg, 20 mmol) was dissolved in a mixture of 57% w/w aqueous HI solution (25.0 ml, 190 mmol) and 50% aqueous H$_3$PO$_2$ (3.4 ml, 15.5 mmol), by heating to boiling under constant magnetic stirring for about 5 min, forming a bright yellow solution. Subsequent addition of solid CH$_3$NH$_3$Cl, of $20 \times (N-1)/N$ mmol, to the hot yellow solution initially caused the precipitation of a black powder which rapidly re-dissolved under stirring to yield a clear bright yellow solution. In a separate beaker, $n$-CH$_3$(CH$_2$)$_3$NH$_2$, of 20-(20 × (N−1)/N) mmol, was neutralized with aqueous HI solution (57% w/w, 5 ml, 38 mmol) in an ice bath resulting in a clear, pale yellow solution. Addition of the $n$-CH$_3$(CH$_2$)$_3$NH$_3$I solution to the PbI$_2$ solution initially produced a black precipitate which was subsequently dissolved on heating the combined solution to boiling. The stirring was then discontinued, and the solution was left to cool to room temperature during which time rectangular-shaped crystals started to crystallize. The precipitation was deemed to be complete after about 24 h. The crystals were isolated by suction filtration and thoroughly dried under reduced pressure. The purity of the materials was confirmed by powder diffraction, and absorption and photoluminescence spectroscopy.

**Static characterization**. Microscopic reflection measurements were performed with a Filmetrics F40 thin-film analyser. Transmission spectra were collected using a customized microscope system. SEM experiments were taken with a JEOL 7500 scanning electron microscope. The scalebar of our SEM tool was calibrated using a calibration standard specimen. Single-crystal $\theta$–$2\theta$ X-ray diffraction was performed with an X-ray diffractometer (Bruker D8 Discover).

**Transient reflection measurements**. Femtosecond transient absorption measurements were performed using a 35 fs titanium:sapphire laser operating at 800 nm at a 2 kHz repetition rate. 400-nm pump pulses were generated via frequency-doubling of the 800-nm amplifier output using a BBO crystal. Pump pulses of other wavelengths were generated from an optical parametric amplifier. The pump pulses were reduced in repetition rate down to 1 kHz with a mechanical chopper. Near-infrared probe pulses were produced by focusing a portion of the amplifier output to a 12-mm-thick YAG crystal.

**Band structure calculation**. The electronic band structures were computed using density functional theory as implemented in the Quantum Espresso package[61] with LDA norm-conserving pseudopotentials. A plane wave kinetic energy cut-off of 80 Ry and a $k$-point grid of $4 \times 1 \times 4$ are used in the calculations. All atomic positions are relaxed to forces less than 0.001 Ry Bohr$^{-1}$. The change in bandgap due to a tensile strain of 1% along the $b$ axis is computed for $N = 1$ and 3.

**Data availability**. All relevant data used in the article are available from the authors.

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

## Acknowledgements

The work was performed at the Center for Nanoscale Materials, a U.S. Department of Energy Office of Science User Facility, and supported by the U.S. Department of Energy, Office of Science, under Contract No. DE-AC02-06CH11357. This material is based upon work supported by Laboratory Directed Research and Development (LDRD) funding from Argonne National Laboratory, provided by the Director, Office of Science, of the U.S. Department of Energy under contract DE-AC02-06CH11357. Work at Northwestern University was supported by grant SC0012541 from the US Department of Energy, Office of Science (sample synthesis, processing and structural characterization).

## Author contributions

P.G. and R.D.S. designed the research. P.G. performed all measurements and coarse-grained modeling. C.C.S. and L.M. synthesized the single crystals. S.S. and P.D. carried out the DFT calculations. J.B.K. contributed to the analysis of results. P.G. wrote the manuscript with input from all authors. R.D.S. supervised the project.

## Additional information

**Competing interests:** The authors declare no competing interests.

