## [Peer Review File · Nature Communications]

Reviewers' Comments:

Reviewer #1:

Remarks to the Author:

The authors reported on the generation and propagation of coherent longitudinal acoustic phonons along the cross-plane direction of two-dimensional Ruddlesden-Popper organic-inorganic hybrid perovskites $BA_2MA_n-1Pb_nI_{3n+1}$. They found a significant reduction in both group velocity and propagation length through experiments. They found that these vibrational properties are caused by a large acoustic impedance mismatch between the organic and inorganic sub-lattices based on their coarse-grained model. This paper is very useful for the development of new materials with tunable phonon transport. However, I'd like to ask the authors some further discussion before accepted.

(1) For the minimal coarse-grained bead-spring model developed by the authors, why did authors only consider the vibrations of ions along b-axis (cross-plane direction)?

(2) In the bead-spring model, the distortion of each Pb-I octahedron is ignored. Do the distortions influence the acoustic phonons? The author should explain this in detail.

(3) In Page 7, the authors said "Because the octahedral sheets in 2D-RPs derive from the (110) planes of the tetragonal $MAPbI_3$ (plane spacing is 6.26 Å), v_a should approximately equal v along the [110] direction of the tetragonal phase, equivalently to the [100] direction of the cubic phase.³⁹" This sentence is not clear, why it is equivalent to the [100] direction of the cubic phase?

(4) In Page 8, "This can be attributed to an increasing ionic character of the inorganic layer as n decreases from 6 to 1 (fewer $[(MA)_n-1Pb_nI_{3n+1}]_2^-$ layers to distribute the -2 negative charge)..." Why does the ionic character increase when n decrease? What is the meaning of "... distribute the -2 negative charge"?

Reviewer #2:

Remarks to the Author:

The authors report on experimental measurements and analysis of the optical refractive index of 2D RP (Ruddlesden-Popper) phase associated to a time-domain optical spectroscopy analysis revealing photoinduced acoustic phonons. These observations permit to extract some elastic parameters (out-of plane sound velocity) and to discuss them in view of the relative importance of the organic/inorganic elements in the so-called 2D-RP systems. The authors also show that this 2D system is softer than the traditional 3D counterpart $MAPbI_3$. Although these materials are of important interest for potential applications and has been well characterized from the optical point of view (refractive index measurements), I think that the part describing the photo induced lattice dynamics suffers from many unclear and unsupported discussions.

I have the following comments.

1) Line 56 : The author write : « We find 2D-RPs exhibit significantly lower group velocities compared to the 3D counterpart, $MAPbI_3$, ». But in the dispersion curve shown in the SI (figure S17) increasing the parameter n , leads to a nearly unchanged of the slope dw/dq in the Brillouin zone center (acoustic branches , right part of S17)? Is there any contradiction or something not well presented on this figure ?

2) Line 83 : "through electron-phonon coupling" ? This part is unclear. Do the authors have an idea of the mechanisms as discussed in the literature (Thomsen et al Phys Rev B 1986, Gusev Laser Optoacoustics, AIP-NewYork 1992, Ruello et al Ultrasonics 2015)

This is important in view of the possible acoustic phonon damping. (see also points 5 and 6)

3) The authors use separate measurements to evaluate first k_a (13-21 N.m⁻¹) in order to get an estimate of k_b (associated to an numerical model to produce some theoretical data). The error/dispersion of k_a is pretty high and much larger (13-21 N.m⁻¹) than the extracted k_b value (around 1 N.m⁻¹). So it is not convincing to announce such a value within that context (large discrepancies of k_a). It is not reasonable. What would be k_b if they fix k_a to 13 N.m⁻¹ ? Many assumptions are done regarding the equivalence of [110] and [100] directions in cubic and tetragonal phase that appear rather speculative and contribute additionally to the general uncertainty of the estimate of k_b .

4) The authors claim that they can extract the bulk modulus from the out-of plane (cross-plane) sound velocity measurement. This is not correct. They can extract the elastic modulus associated to this zz strain/stress tensor. To get the bulk modulus they need to know the Poisson coefficient (see Thomsen et al Phys Rev B 1986). This has to be corrected

5) Line 200 : the authors observe an increase of the CLAP with decreasing n. The authors explain this effect is an increase of the displacement of the Pb-I octahedra. The authors rule out a detection mechanism. This is not strongly supported. To prove it, the authors must show that while varying the n parameter, the photoelastic coefficients (linked to the deformation potential parameter and the derivative as a function of probe energy of the real and imaginary part of the refractive index) that play a crucial role in the magnitude of detected signal (see Thomsen et al Phys Rev B 1986) remain unchanged, which is currently unknown, I think ?

6) The authors implicitly consider the generation mechanism is the thermoelastic effect (line 253 in the SI) while they do not prove it. This claim needs to be supported more since the authors do not discuss possible other generation mechanisms (increase of deformation potential parameter or thermoelastic effect – see Thomsen et al Phys Rev B 1986, Ruello et al Ultrasonics 2015, Gusev Laser Optoacoustics, AIP-NewYork 1992). A slight n dependence of the band gap (the authors say that E_g varies with n in line 123) can also lead to a n dependence of the deformation potential as know in confined materials such as quantum dots, well or ultrathin films (Mittleman, D. M. et al. Phys. Rev. B 49, 14435 (1994), Allan, G. & Delerue, C. Phys. Rev. B 70, 245321 (2004)). Such confinement could change the efficiency of CAP generation/detection through deformation potential mechanism (see different example in the literature Devos et al Phys Rev Lett 98, 207402 2007, Weis et al Sci. Report 7 (1), 13782 2017. This possibility is also in line with what the authors write in the introduction « with natural quantum-well like electronic structures » (line 41).

7) The authors mention that the damping changes versus n. The damping of the signal can come from intrinsic anharmonic effect but there is in general a part due to a partial penetration of the probe beam (imaginary part of the refractive index). Did the author estimate it ?

8) In the manuscript as well as in the conclusion, the authors say that the reduction of the group velocity is due to a impedance mismatch. As far as I know, impedance mismatch only govern the amplitude of acoustic wave reflection/transmission ($R=(Z1-Z2)/(Z1+Z2)$) and not the sound velocity. This unclear part has to be corrected.

9) In conclusion, the authors finally say that this 2D-RP system has phononic properties. I think no evidence of this (flowing of phonon branches) is reported here.

Summary :

The manuscript reports on several experimental observations of light-induced coherent acoustic phonon in 2D RP single crystal and on a full characterization of the optical refractive index. While I have no specific criticism regarding the optical characterization, I think there are some incorrect statements in the manuscript, the discussion is not enough supported and some conclusions appear as too speculative in the present form, specifically concerning the photo induced lattice dynamics. As a consequence, I cannot recommend unfortunately the publication in Nature

Communications.

Reviewer #3:

Remarks to the Author:

The manuscript by Guo et al. describes a very nice investigation on the coherent longitudinal acoustic phonons in 2D layered perovskites with $n=1$ to $n=6$ employing ultrafast pump-probe spectroscopy. These experiments provide direct measurements of cross-plane group velocities and coherence time for acoustic phonons. The results are important and of interest to the community, and I support the publication of this manuscript in Nature Communications.

There are a couple of results that puzzle me, and I hope the authors can clarify.

1. In Figure 4b, the acoustic phonon velocity is higher for the $n=1$ than the $n=2$ sample. Could the substrate play any role in phonon velocity?
2. In Figure 5c, the coherence time remains essentially constant from $n=1$ to $n=6$ and they are all ~ 6 time shorter than 3D MAPbI₃. Could the authors elaborate on the seemingly independence of coherence time on n ?

Reviewer #1 (Remarks to the Author):

The authors reported on the generation and propagation of coherent longitudinal acoustic phonons along the cross-plane direction of two-dimensional Ruddlesden-Popper organic-inorganic hybrid perovskites $BA_2MA_{n-1}Pb_nI_{3n+1}$. They found a significant reduction in both group velocity and propagation length through experiments. They found that these vibrational properties are caused by a large acoustic impedance mismatch between the organic and inorganic sub-lattices based on their coarse-grained model. This paper is very useful for the development of new materials with tunable phonon transport. However, I'd like to ask the authors some further discussion before accepted.

(1) For the minimal coarse-grained bead-spring model developed by the authors, why did authors only consider the vibrations of ions along b-axis (cross-plane direction)?

Our measurements only probed acoustic phonon propagations along the cross-plane direction, but not along the in-plane directions (i.e., along the perovskite sheets). In this case, the atomic motions along the cross-plane direction are decoupled from motions along the in-plane directions. As a result, the minimal, one-dimensional coarse-grained model is sufficient to capture the phonon characteristics and elastic modulus along the cross-plane direction, which presents large acoustic impedance mismatch. To clarify this point, we have added sentences “Such a 1D model is sufficient to capture the essence of the long-wavelength phonon propagation along the cross-plane direction, which is the experimentally measured quantity” in the revised manuscript. Further study of the acoustic phonon propagating along the in-plane directions, as we pointed out in the paper, may be enabled by ultrafast pump-probe *microscopy*, which is more suitable for measurements on the small cross-sectional area (i.e., the surface of the crystals along the *ac*-plane) of the layered perovskites, but is not available to us at this time.

(2) In the bead-spring model, the distortion of each Pb-I octahedron is ignored. Do the distortions influence the acoustic phonons? The author should explain this in detail.

In the bead-spring model, each layer of Pb-I perovskite sheet is modeled as one bead. As a result, the effect of Pb-I octahedra distortions on the acoustic phonon velocity along the cross-plane direction is implicitly incorporated into the effective spring constant (k_a) that connects the beads. We expect that the influence from the distortions on the velocity does exist. Our estimation of k_a is based on the velocity of 3D MAPbI₃, in which the distortion of Pb-I octahedra may differ from

that in 2D perovskites. However, as shown in an earlier paper (*ACS Energy Lett.*, **2017**, *2*, 2463-2469; Fig. 3f), the distortion of the Pb-I octahedra, which is expected to vary with temperature within the tetragonal phase (e.g., see *Sci. Rep.*, **2016**, *6*, 35686), only leads to ~ 10% change in the velocity as the temperature is changed from 300 K to 160 K. The 10% change partly contributed by octahedra distortions is significantly less than the uncertainty of estimated k_a values that vary from $13 \text{ N}\cdot\text{m}^{-1}$ to $22 \text{ N}\cdot\text{m}^{-1}$ as shown in Fig. 4d. As a result, the difference in Pb-I octahedra distortion is expected to lead to only small corrections, but not change our main conclusion about the one-order-of-magnitude difference in k_a and k_b , where k_b is the spring constant for the spring connecting two Pb-I layers separated by an organic spacer layer. In the revised manuscript, we have added these sentences, “We note that the effect of Pb-I octahedra distortions on the acoustic phonon velocity along the cross-plane direction is implicitly incorporated into the effective spring constant (k_a) that connects the beads. By fixing the value of k_a , the differences in octahedral rotations in $N=1$ to 6 are implicitly neglected, because we found⁴⁰ that octahedral rotations, which occurred in MAPbI₃ as temperature is varied from 300 K to 160 K, only results in a minor change of velocity within 10%.”, to elucidate this point in detail.

(3) In Page 7, the authors said “Because the octahedral sheets in 2D-RPs derive from the (110) planes of the tetragonal MAPbI₃ (plane spacing is 6.26 Å), v_a should approximately equal v along the [110] direction of the tetragonal phase, equivalently to the [100] direction of the cubic phase.³⁹” This sentence is not clear, why it is equivalent to the [100] direction of the cubic phase?

In the revised manuscript, we added Supplementary Fig. 14 to illustrate the crystallographic relationships between the 2D-RPs and 3D MAPbI₃. Note that, for the tetragonal phase of MAPbI₃, the octahedral layers along the [100] direction are stacked in a staggered manner, whereas those along the [110] direction are stacked in an eclipsed manner; the latter is similar to the stacking configuration in 2D-RPs (Fig. 1a). In the tetragonal-to-cubic phase transition of MAPbI₃, the [110] direction ([100] direction) of the tetragonal phase evolve into the [100] direction ([110] direction) of the cubic phase, only that the octahedra are tilted during the phase transition. Besides adding the supplementary figure, we have also improved our description in the main text by adding these sentences, “in the sense that the perovskite layers in both cases are stacked in an eclipsed manner (see Supplementary Fig. 14)”.

(4) In Page 8, “This can be attributed to an increasing ionic character of the inorganic layer as n decreases from 6 to 1 (fewer $[(MA)_{n-1}Pb_nI_{3n+1}]^{2-}$ layers to distribute the -2 negative charge)...” Why does the ionic character increase when n decrease? What is the meaning of “... distribute the -2 negative charge”?

In the 2D-RP series investigated in this work, each BA^+ organic spacer cation always carries one positive charge. Because the chemical composition is $(BA)_2MA_{n-1}Pb_nI_{3n+1}$, the -2 negative charge is distributed by $[(MA)_{n-1}Pb_nI_{3n+1}]^{2-}$, which, depending on the n value (which is the number of perovskite layers), leads to a different charge density per perovskite layer. This is what we meant in our original manuscript “... distribute the -2 negative charge”. In the revised manuscript, we further clarified this point with the following new sentences, “This can be attributed to an increasing ionic character of the inorganic layer as N decreases from 6 to 1. In particular, because N layers of perovskite, with the chemical formula $[(MA)_{N-1}Pb_NI_{3N+1}]^{2-}$, always carry -2 charges to compensate the +2 charges carried by $(BA)_2^{2+}$, each perovskite layer on average distributes $-2/N$ charge.”.

Reviewer #2 (Remarks to the Author):

The authors report on experimental measurements and analysis of the optical refractive index of 2D RP (Ruddlesden-Popper) phase associated to a time-domain optical spectroscopy analysis revealing photoinduced acoustic phonons. These observations permit to extract some elastic parameters (out-of plane sound velocity) and to discuss them in view of the relative importance of the organic/inorganic elements in the so-called 2D-RP systems. The authors also show that this 2D system is softer than the traditional 3D counterpart MAPbI₃. Although these materials are of important interest for potential applications and has been well characterized from the optical point of view (refractive index measurements), I think that the part describing the photo induced lattice dynamics suffers from many unclear and unsupported discussions.

We thank the Reviewer for the careful reading of our manuscript. We'd like to highlight that, in addition to the refractive index characterization, our report of the acoustic phonon velocity, which is the main emphasis of the work, is of great accuracy. Other methods (such as pulse echo) are not compatible with the small dimensions & irregular shapes of the materials, wherein acoustic waves are also strongly attenuated. In addition, our measurements do not require the

guess of a Poisson's ratio, which was typical in other studies wherein the velocity was calculated from Young's modulus that is obtained from nanoindentation experiments (e.g., *Nat. Mater.*, **2016**, *16*, 83-88). To address the concerns regarding photo-induced lattice dynamics, we have performed additional experiments, as well as reworded some of our discussions.

I have the following comments.

1) Line 56: The authors write: We find 2D-RPs exhibit significantly lower group velocities compared to the 3D counterpart, MAPbI₃. But in the dispersion curve shown in the SI (figure S17) increasing the parameter n , leads to a nearly unchanged of the slope dw/dq in the Brillouin zone center (acoustic branches, right part of S17)? Is there any contradiction or something not well presented on this figure?

We note that in Fig. S17 of the original manuscript (which is Supplementary Fig. 20 in the revised version), we only showed dispersion curves for $N=1$ to $N=6$, but not for $N=\infty$ (which corresponds to MAPbI₃). It is accurate that the velocity of MAPbI₃ is very different from the layered counterparts, but as plotted in Fig. 4b, the velocities for $N=1$ to $N=6$ do not have significant differences (which is an important finding of our manuscript). This phenomenon can be understood when considering each unit cell of 2D perovskites as a number of springs connected in series, the modulus is primarily determined by the weakest spring. As such, as soon as organic spacers (i.e., soft springs) are inserted between the perovskite layers, the modulus is significantly decreased as compared to the case without organic spacers. As a result, the difference between $N=1$ and $N=6$ is not large, and we believe Supplementary Fig. 20 and Line 56 are both correct.

2) Line 83: "through electron-phonon coupling"? This part is unclear. Do the authors have an idea of the mechanisms as discussed in the literature (Thomsen et al Phys Rev B 1986, Gusev Laser Optoacoustics, AIP-NewYork 1992, Ruello et al Ultrasonics 2015) This is important in view of the possible acoustic phonon damping. (see also points 5 and 6).

In order to address this important question, we have performed additional measurements with carefully varied pump wavelengths and fluences. More details regarding this question can be found in answers for questions 6.

3) The authors use separate measurements to evaluate first k_a (13-21 $\text{N}\cdot\text{m}^{-1}$) in order to get an estimate of k_b (associated to a numerical model to produce some theoretical data). The error/dispersion of k_a is pretty high and much larger (13-21 $\text{N}\cdot\text{m}^{-1}$) than the extracted k_b value (around 1 $\text{N}\cdot\text{m}^{-1}$). So, it is not convincing to announce such a value within that context (large discrepancies of k_a). It is not reasonable. What would be k_b if they fix k_a to 13 $\text{N}\cdot\text{m}^{-1}$? Many assumptions are done regarding the equivalence of [110] and [100] directions in cubic and tetragonal phase that appear rather speculative and contribute additionally to the general uncertainty of the estimate of k_b .

We would like to point out that Fig. 4d in the original manuscript can exactly serve to answer these questions. In Fig. 4d, we showed how k_b changes with k_a for each member of the 2D-RPs, so as to yield calculated velocities that match with the measured ones as shown in Fig. 4b. We found that even with k_a varied from 13 $\text{N}\cdot\text{m}^{-1}$ to 21 $\text{N}\cdot\text{m}^{-1}$, variation of k_b is only within 10%. This fact can again be understood (partly discussed in answer to question 1), by the fact that the modulus in 2D-RPs is primarily determined by two adjacent layers with the weakest interactions, which is k_b in the present case. Therefore, we believe our conclusion that k_a and k_b differ by an order of magnitude is still correct. To answer the question “What would be k_b if they fix k_a to 13 $\text{N}\cdot\text{m}^{-1}$ ”, we’d like to note that it was possible to read from Fig. 4d the k_b values when fixing k_a to 13 $\text{N}\cdot\text{m}^{-1}$, which is at the lower limit of the plot. Because 2D-RPs exhibit much slower velocities compared to the 3D case, k_b has to be an order of magnitude smaller than k_a . It is exactly this one order-of-magnitude difference between k_a and k_b , that makes k_a no longer important compared to k_b , in the sense that variation of k_a can be large without significantly changing the final velocity. In the revised manuscript, we added the sentence “For any given member, the small variation of k_b (less than 10%) relative to the variation of k_a (13~22 $\text{N}\cdot\text{m}^{-1}$) stems from the order-of-magnitude difference between k_b and k_a ; such difference dictates that the CLAP velocity is mainly determined by k_b , while a relatively large variation of k_a does not significantly alter the velocity.” to clearly point this out and hopefully satisfy the Reviewer. This behavior also ensures the conversion between the [110] and [100] directions of the tetragonal and cubic phases, which cannot result in order-of-magnitude change of k_a . More discussion regarding the equivalence of the [110] and [100] directions of the tetragonal and cubic phases were also added in our answer to question 3 of the first Reviewer.

Intuitively, we know that springs connected in series is analogous to resistors connected in parallel. In the latter case, the total resistance of two resistors is primarily determined by the resistor with the smaller resistance, and changing the resistance of the resistor with the larger resistance does not significantly influence the total resistance.

4) The authors claim that they can extract the bulk modulus from the out-of plane (cross-plane) sound velocity measurement. This is not correct. They can extract the elastic modulus associated to this zz strain/stress tensor. To get the bulk modulus they need to know the Poisson coefficient (see Thomsen et al Phys Rev B 1986). This has to be corrected.

We thank the Reviewer for pointing this out, and have corrected this error in the revised manuscript.

5) Line 200: the authors observe an increase of the CLAP with decreasing n . The authors explain this effect is an increase of the displacement of the Pb-I octahedra. The authors rule out a detection mechanism. This is not strongly supported. To prove it, the authors must show that while varying the n parameter, the photoelastic coefficients (linked to the deformation potential parameter and the derivative as a function of probe energy of the real and imaginary part of the refractive index) that play a crucial role in the magnitude of detected signal (see Thomsen et al Phys Rev B 1986) remain unchanged, which is currently unknown, I think?

We agree with the Reviewer that a fully quantitative assessment of the strain induced reflection of the probe requires knowledge of the photoelastic (PE) coefficients. A proper measurement of the PE coefficient requires transient X-ray experiments (or similar techniques) to fully map out the strain distribution as a function of CAP frequency. Such experiments are beyond the scope of our paper, in which we try to systematically investigate the effect of organic spacers on the acoustic phonon transport in these unique material systems. As such, we have shortened our discussion related to this point, and also reworded the sentence to read as follows: “A similar dependence of $\Delta\langle|u(t)|\rangle$ on N is observed, which may contribute to the observed N -dependent oscillation amplitude”, and “Full assessment of the oscillation amplitude requires knowledge of the photoelastic coefficients (i.e., strain dependent refractive index change), which is beyond the scope of the present work”.

6) The authors implicitly consider the generation mechanism is the thermoelastic effect (line 253 in the SI) while they do not prove it. This claim needs to be supported more since the authors do

not discuss possible other generation mechanisms (increase of deformation potential parameter or thermoelastic effect – see Thomsen et al Phys Rev B 1986, Ruello et al Ultrasonics 2015, Gusev Laser Optoacoustics, AIP-NewYork 1992). A slight n dependence of the band gap (the authors say that E_g varies with n in line 123) can also lead to a n dependence of the deformation potential as known in confined materials such as quantum dots, well or ultrathin films (Mittleman, D. M. et al. Phys. Rev. B 49, 14435 (1994), Allan, G. & Delerue, C. Phys. Rev. B 70, 245321 (2004)). Such confinement could change the efficiency of CAP generation/detection through deformation potential mechanism (see different example in the literature Devos et al Phys Rev Lett 98, 207402 2007, Weis et al Sci. Report 7 (1), 13782 2017. This possibility is also in line with what the authors write in the introduction « with natural quantum-well like electronic structures » (line 41).

We thank the Reviewer for this insightful comment. CAPs can be launched *via* both the thermoelastic and the deformation potential effects. Thermoelastic effect relates impulsive heating of the lattice by hot carriers; these hot carriers (including both electrons and holes) quickly relax to the exciton energy through equilibration with the lattice, which causes a lattice thermal expansion and with it CAPs. As to the deformation potential mechanism, strain is induced by the modification in energy of the electronic distribution, which in our case involves electrons and holes created by the above-exciton-gap pump.

In our original manuscript, we implicitly assigned the generation of CAPs primarily to the thermoelastic effect. This assignment stemmed from our observation that a near exciton-gap pump excitation was not able to drive the CAPs (which we did not include in the original paper). As a result, we used 400 nm pump, which is far above the exciton energies for all the members, to examine all the compositions. For the revision of our manuscript, we performed additional transient reflection measurements, where we varied the pump wavelengths for two representative compositions ($n=2$ from the low members and $n=4$ from the high members). These results are now incorporated and presented as Supplementary Figs. 22, 23 and 24. We found that for both compositions, a slightly above-exciton-gap pump (570 nm for $n=2$ and 640 nm for $n=4$), which could generate electron-hole pairs (evident from the induced absorption with amplitude comparable to that generated by 400 nm pump), did not yield apparent oscillations of the probe, which strongly suggests that thermoelastic effect plays a dominant role in launching the CAPs in this system.

We also noted that a different penetration depth of the pump may alter the frequency distribution of the CAPs (Nat. Commun., 8, 14398) and with it their detectability. Although a literature report of the above-bandgap index of refraction is currently not available, we learned from this paper (arXiv:1710.07653, Fig. 3) that $N=2$ has different absorption coefficients at 500 nm and 570 nm. Note that, although 500 nm pump yields stronger probe oscillations compared to 570 nm pump (which is consistent with a thermoelastic effect dominated CAP generation mechanism), both 500 nm and 570 nm pump wavelengths show substantially weaker CAP oscillations compared to the 400-nm pump excitation, thereby ruling out the possibility that different penetration depths lead to the observed different strengths of the probe oscillations.

Based on the above reasoning, we believe that our qualitative conclusion derived from the minimal bead-spring model, that the mean bead displacement increases as N decreases, is indeed correct. In the revised manuscript, we added the discussions, “Pump wavelength and fluence dependent measurements on $N=2$ (strongly quantum confined) and $N=4$ (weakly quantum confined), as shown in Supplementary Figs. 22 to 24, demonstrate that in both cases, the CLAP signatures in the $\Delta R/R$ spectral maps can only be obtained *via* above-bandgap pumping. Therefore, the thermoelastic effect is mainly responsible for the CLAP signatures in the transient reflection spectral maps.”, to point to the added plots in the Supporting Info regarding the generation mechanism of the CAPs. Similar changes was made in the Supplementary Note 3 to refer to these new results.

7) The authors mention that the damping changes versus n . The damping of the signal can come from intrinsic anharmonic effect but there is in general a part due to a partial penetration of the probe beam (imaginary part of the refractive index). Did the author estimate it?

We would like to point out that the penetration depth of the probe is not the limiting factor in the presented work. Our transient reflection measurements were performed for the *below-bandgap* range, with nearly zero optical absorption (i.e., the imaginary part of the index vanishes), for both the 2D and 3D perovskites. The absorption coefficients for 2D perovskites are shown, for example, in reference 10. This point has been stated in the original manuscript in Line 89 and Lines 97-98. Further demonstration of complete penetration of the probe beam was manifested in Fig. S12, where the transmission was very large (and went out-of-scale). In the revised manuscript, we have now re-plotted Supplementary Fig. S12 by dividing the transmission by a

factor of 3. It is now clear that T+R is nearly unity, indicating a negligible optical absorption. According to our answer to the 1st question by Reviewer 3, the acoustic phonon attenuation takes place within a few microns of propagation length, which is thinner than the entire thickness of the flakes, further ensuring that the observed attenuation does not arise from optical absorption.

For clarification, in the revised manuscript we added the sentences “Note that measurements in the below-bandgap range with negligible optical absorption (Supplementary Fig. 12) ensure that optical absorption does not contribute to the observed acoustic phonon attenuation”.

8) In the manuscript as well as in the conclusion, the authors say that the reduction of the group velocity is due to an impedance mismatch. As far as I know, impedance mismatch only governs the amplitude of acoustic wave reflection/transmission ($R=(Z_1-Z_2)/(Z_1+Z_2)$) and not the sound velocity. This unclear part has to be corrected.

The acoustic impedance (Z) is defined as $Z=\rho v$, where ρ is the mass density and v is the speed of sound. We claimed that in a superlattice composed of two different materials (here being the inorganic perovskite layers and organic spacer layers), an impedance mismatch can result in the reduction of acoustic phonon velocity of the superlattice.

Such claim has literature precedent. For example, in this paper (*Phys. Rev. B.*, **1988**, 38, 1427), the sound speed of a superlattice (denoted as v) can be calculated analytically as $v = D \left\{ \left(\frac{d_A}{v_A} \right)^2 + \left(\frac{d_B}{v_B} \right)^2 + \frac{1+\delta^2}{\delta} \frac{d_A d_B}{v_A v_B} \right\}^{-1/2}$, where d_A (d_B) is the thickness of material A (B) in each unit cell, $D = d_A + d_B$ is the total thickness of the unit cell, v_A (v_B) is the sound speed of material A (B), and $\delta = \frac{\rho_A v_A}{\rho_B v_B}$ is the ratio of the acoustic impedance. For δ much larger than unity, we have $\frac{1+\delta^2}{\delta} \rightarrow \delta$, hence the term $\frac{1+\delta^2}{\delta} \frac{d_A d_B}{v_A v_B}$ can become very large in comparison to either the $\left(\frac{d_B}{v_B} \right)^2$ or the $\left(\frac{d_A}{v_A} \right)^2$ term, which will result in v that is smaller than either v_A or v_B . Note that in this model

the displacement and stress of material A and B at the interface are continuous.

We could gain a similar conclusion using a simple bead-spring model. The unit cell of the superlattice is shown on the

left. The sound speed can be calculated from $v = (E/\rho)^{1/2}$, where E is the modulus and ρ is the mass density, both calculable from the parameters shown in the figure. For a chain composed of just blue beads (material A), we have $\rho_A = m_A/(Sd_A)$ and $v_A = (k_A/m_A)^{1/2}d_A$, here S is the cross-sectional area of the unit cell. Similarly, $\rho_B = m_B/(Sd_B)$ and $v_B = (k_B/m_B)^{1/2}d_B$ for material B. For the superlattice, we have $\rho = (2m_A + 2m_B)/[S(d_A + d_B + 2d_{AB})]$, and

$$v = \left[\left(\frac{1}{\left(\frac{1}{k_A} + \frac{1}{k_B} + \frac{2}{k_{AB}} \right) (2m_A + 2m_B)} \right) \right]^{1/2} (d_A + d_B + 2d_{AB}).$$

Assume that the displacement and stress of material A and B at the interface are continuous (i.e., k_{AB} is large and d_{AB} is small), we have

$$v \cong \left[\left(\frac{1}{\left(\frac{1}{k_A} + \frac{1}{k_B} \right) (2m_A + 2m_B)} \right) \right]^{1/2} (d_A + d_B).$$

A large impedance mismatch means that $\rho_A v_A = m_A(k_A/m_A)^{1/2}/S$ is much larger than $\rho_B v_B = m_B(k_B/m_B)^{1/2}/S$, or equivalently, $k_A m_A \gg k_B m_B$. In this case, the term $\frac{1}{\left(\frac{1}{k_A} + \frac{1}{k_B} \right) (2m_A + 2m_B)} = \frac{k_A k_B}{(k_A + k_B)(2m_A + 2m_B)} \cong \frac{k_A k_B}{2k_A m_A} = \frac{k_B}{2m_A}$, and

$$v \cong \left(\frac{k_B}{2m_A} \right)^{1/2} (d_A + d_B),$$

which can be smaller than $v_B = (k_B/m_B)^{1/2}d_B$, when $\left(\frac{d_A + d_B}{d_B} \right)^2 < \frac{2m_A}{m_B}$. Note that the acoustic impedance does not involve the d_A and d_B terms, so $\left(\frac{d_A + d_B}{d_B} \right)^2 < \frac{2m_A}{m_B}$

could be realized without violating $\rho_A v_A \gg \rho_B v_B$.

We see that, both models described above can result in reduced sound speed in the superlattice, in comparison to the individual velocities, if the impedance mismatch is large, and provided that displacement and stress of material A and B at the interface are continuous. Even if the displacement and stress of the two materials at the interface were not strictly continuous, the composite will still exhibit a velocity that *approaches the lower velocity of the two individual components*, if the impedance mismatch is large. In our case, because the organic component (butylammonium) has much lower velocity than the inorganic component (MAPbI₃), the 2D-RPs exhibit velocities as low as ~1425 m/s, which approaches the velocity of liquid phase of butylammonium (1250 m/s) with randomized orientation of the molecules. However, we do note that in the present case, because the organic spacers are arranged in a periodic fashion that is very different than the liquid phase, it is hard to assign a velocity for the organic layer. But usually a well-ordered structure should exhibit a higher sound velocity (which has been seen for

liquid crystals, as we pointed out in the manuscript), which means that the 2D-RPs really approach the lower velocity of the two highly impedance mismatched sub-lattices.

Based on the above discussion, together with the fact that the system in the present study is one of the highest impedance mismatched systems that differs from all-inorganic superlattices (e.g., Si-Ge superlattices), we believe our claim, that reduction of the group velocity is due to an impedance mismatch, is still correct.

9) In conclusion, the authors finally say that this 2D-RP system has phononic properties. I think no evidence of this (flowing of phonon branches) is reported here.

We agree with the Reviewer that our manuscript does not report phononic properties regarding to flow and/or folding of phonon branches, which are exciting areas that may warrant further studies. In the revised manuscript, we have changed the sentence to “... feature a new class of materials with tunable acoustic phonon transport characteristics...”

Summary: The manuscript reports on several experimental observations of light-induced coherent acoustic phonon in 2D RP single crystal and on a full characterization of the optical refractive index. While I have no specific criticism regarding the optical characterization, I think there are some incorrect statements in the manuscript, the discussion is not enough supported and some conclusions appear as too speculative in the present form, specifically concerning the photo induced lattice dynamics. As a consequence, I cannot recommend unfortunately the publication in Nature Communications.

We hope that our additional experiments and discussion can sufficiently address the concerns of the Reviewer. In addition, we have added insightful results on one more composition, (HA)PbI₄, which we think can still further strengthen the paper.

Reviewer #3 (Remarks to the Author):

The manuscript by Guo et al. describes a very nice investigation on the coherent longitudinal acoustic phonons in 2D layered perovskites with n=1 to n=6 employing ultrafast pump-probe spectroscopy. These experiments provide direct measurements of cross-plane group velocities and coherence time for acoustic phonons. The results are important and of interest to the community, and I support the publication of this manuscript in Nature Communications. There are a couple of results that puzzle me, and I hope the authors can clarify.

1. In Figure 4b, the acoustic phonon velocity is higher for the $n=1$ than the $n=2$ sample. Could the substrate play any role in phonon velocity?

The samples we examined with transient reflection measurements are thicker than tens of microns, whereas the sample depth relevant for the acoustic phonon propagation is much thinner. Specifically, in Fig. 2, each period (in time) of the oscillation corresponds to a phonon propagation distance equal to half the wavelength of light in the materials. This arises because two consecutive peaks in the oscillation (both corresponding to constructive interference) have a phase shift of 2π . For all the members shown in Fig. 2, we probed at most 5 periods of oscillations, which indicate a propagation distance no longer than several microns. As a result, only the very top, several microns of the sample contribute to the probed acoustic phonons, and the substrate does not play a role in our measurements. Also, in fact, our samples are not directly grown on any substrates, but are physically attached to a substrate using double-sided carbon tape. To clarify this point, in the revised manuscript we have added sentences “Each period (in time) of the oscillation corresponds to a phonon propagation distance equal to half the wavelength of light in 2D-RPs. As a result, the propagation distances of the CLAPs shown in Fig. 2 are on the order of a few microns.”.

2. In Figure 5c, the coherence time remains essentially constant from $n=1$ to $n=6$ and they are all ~ 6 time shorter than 3D MAPbI₃. Could the authors elaborate on the seemingly independence of coherence time on n ?

We thank the Reviewer for this insightful question. We think that the seeming independence of the coherence time on n may possibly arise from three aspects. (i) It is known that for layered perovskites, higher members are in general more difficult to crystalize than lower members, because the phase stability decreases with an increasing n . Note that most published studies on layered perovskites have focused on lower members, possibly partially resulting from this fact. Although our samples are single phases as identified from XRD patterns, it is possible that higher members have more defects than lower members. For example, a recent paper (*Science*, **2017**, 355, 1288-1292) shows that layered perovskites with $N>2$ exhibit lower-energy edge states, which are possibly attributable to defects. The defects that are more likely to be present in higher members may introduce additional scattering of the acoustic phonons. (ii) Although lower N members have higher density of interfaces, the wave-like nature of acoustic phonons can be

more strongly manifested at higher interfacial densities, as discussed in several seminal papers (*Nat. Mater.*, **2014**, *13*, 168-172; *Science*, **2012**, *338*, 936-939). The wave-like nature and with its coherent phonon scattering (as compared to diffuse scattering) may contribute to a lower scattering efficiency (per inorganic-organic interface) of the CLAPs in lower members. (iii) Our results show that lower members have stronger bonding strength at the organic interfaces (see Fig. 4b and 4d). In other words, larger spring constants have to be assumed for lower members (as compared to those for higher members) in order to match the calculated CLAP velocities with the experimental values. Such observation may suggest that, the dynamic disorder in lower members is less pronounced than that in higher members. Relatedly, our results on (HA)PbI₄ shows that, a higher CLAP velocity is accompanied by a weaker CLAP attenuation (see Fig. 5f).

To illustrate these points, in the revised manuscript we have added the sentences “For $N=1\sim 6$, a seemingly independence of the coherence time on N may arise from (i) Larger defect densities in higher N members which may contribute to additional acoustic phonon scattering¹⁴; (ii) A stronger manifestation of the wave-like nature of CLAPs in lower N members with a higher interfacial density¹⁹; (iii) Lower N member exhibit stronger bonding strength at the organic interface (Fig. 4d), which possibly results in a less pronounced dynamic disorder. More quantitative understanding may be aided by additional measurements including the phonon mean free paths^{22,49}”.

Reviewers' Comments:

Reviewer #1:

Remarks to the Author:

I advise that the revised manuscript can be published according to the authors' reply.

Reviewer #2:

Remarks to the Author:

Report on the revised manuscript :

The authors have performed additional experiments and have provided new materials that make the paper clearly more convincing. All the issues have been well addressed in the response letter. This complete work will be of a broad interest for the community of materials science in general and for the communities focusing on hybrid materials and light-induced lattice dynamics in particular.

I recommend now the paper for publication.

Reviewer #3:

Remarks to the Author:

I think the authors have addressed my and other reviewers' comments in the revised manuscript. I support the publication of this manuscript in Nature communications.